# Community Detection in Large-Scale Complex Networks via Structural Entropy Game

## ABSTRACT

Community detection is a critical task in graph theory, social network analysis, and bioinformatics, where communities are defined as clusters of densely interconnected nodes. However, detecting communities in large-scale networks with millions of nodes and billions of edges remains challenging due to the inefficiency and unreliability of existing methods. Moreover, many current approaches are limited to specific graph types, such as unweighted or undirected graphs, reducing their broader applicability. To address these limitations, we propose a novel heuristic community detection algorithm inspired by game theory, termed CoDeSEG, which identifies communities by minimizing the network's two-dimensional (2D) structural entropy. In this potential game model, nodes decide whether to stay or transfer to another community based on a strategy that maximizes a 2D structural entropy utility function. Additionally, we introduce a structural entropy-based node overlapping heuristic to detect overlapping communities. The algorithm operates with near-linear time complexity, enabling efficient community detection in large-scale networks. Experimental results on real-world networks demonstrate that CoDeSEG is the fastest method available and achieves state-of-the-art performance in overlapping normalized mutual information (ONMI) and F1 score.

## KEYWORDS

Community Detection, Structural Entropy, Potential Games, Large-scale Networks.

**ACM Reference Format:**

Anonymous Author(s). 2025. Community Detection in Large-Scale Complex Networks via Structural Entropy Game. In *Proceedings of the ACM Web Conference 2025 (WWW '25), April 28–May 2, 2025, Sydney, Australia.* ACM, New York, NY, USA, 12 pages. https://doi.org/10.1145/nnnnnnn.nnnnnnn

## 1 INTRODUCTION

Community refers to a set of closely related nodes within a network, also known as a cluster or module in literature [14, 16]. Community detection is a task that reveals fundamental structural information within real-world networks, providing valuable insights by identifying tightly knit subgroups. In drug discovery, for instance, detecting protein functional groups facilitates the identification of novel, valuable proteins [32]. In social event detection, analyzing message groups within social streams helps to understand the development

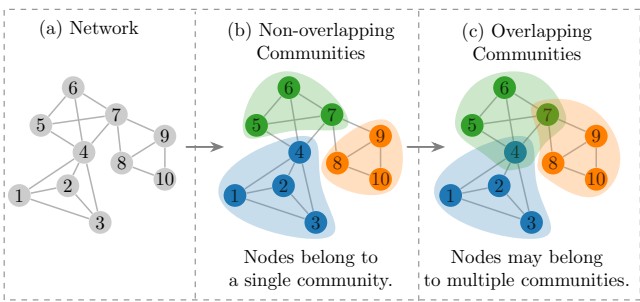

**Figure 1: Illustration of non-overlapping and overlapping community structures in a network.**

trends of events and analyze public sentiment [8]. Community detection also plays a role in recent retrieval-augmented generation (RAG) applications, like GraphRAG [12]. Furthermore, community detection has extensive applications across various domains, including recommender systems [4], medicine [3], biomedical research [33, 40], social networks [15, 38], and more.

As illustrated in Figure 1(b), most early research on community detection has focused on disjoint clusters, where each node belongs to a single community, and there is no overlap between communities [7, 14, 39, 43, 45]. However, nodes often participate in multiple communities in many real-world applications (as depicted in Figure 1(c)), sparking a growing interest in detecting overlapping communities [22, 37, 48]. Overlapping community detection typically entails higher computational costs and time overhead than disjoint community detection. In the past two decades, numerous algorithms for overlapping community detection have been proposed, including those based on modularity [10], label propagation [28, 49], seed expansion [19, 47], non-negative matrix factorization [50], spectral clustering [46]. However, existing overlapping community detection methods [9, 17, 24, 50] are not capable of large-scale networks with millions of nodes and billions of edges. These algorithms often require several days, or even longer, to achieve satisfactory results.

Many well-established and widely-used community detection methods for large-scale networks are typically limited to specific types of graphs, such as unweighted or undirected graphs, thereby restricting their applicability. For instance, algorithms like Bigclam [50] and SLPA [49] detect overlapping communities in unweighted and undirected graphs. Methods like Louvain [7], Leiden [43], and LPA [39] focus on detecting non-overlapping communities in undirected graphs. Detecting overlapping communities in weighted, directed, large-scale networks remains a significant challenge.

In recent years, deep learning-based community detection models have achieved promising results by learning node embeddings and detecting communities through node clustering or classification. However, due to the learning and encoding processes of deep

models, these methods demonstrate inefficiencies when applied to large-scale networks[20, 42].

To tackle these challenges, we propose a novel algorithm named CoDeSEG (**Co**mmunity **De**tection via **S**tructural **E**ntropy **G**ame) for detecting overlapping communities in large-scale complex networks. The proposed algorithm follows the game-theoretic inspired community detection framework [9], named as **community formation game**. In the game, nodes join or leave communities by maximizing their utility function. The Nash equilibrium of the game directly corresponds to the network's community structure, with each node's community memberships at equilibrium serving as the output of the community detection algorithm.

The community-formation-game-based algorithm shows its effectiveness and efficiency in large-scale networks. Lyu et al. propose the FOX [31] algorithm, which measures the closeness between nodes and communities by approximating the number of triangles in communities. Ferdowsi et al. introduce a two-phase non-cooperative game model for community detection, where non-overlapping communities are first identified using a local interaction utility function, followed by identifying overlapping nodes based on the payoffs derived from community memberships [13].

In contrast to these methods, we define the potential function as the 2-dimensional structural entropy (2D SE) [26] of the network. We further derive an efficient node utility function from the potential function, which can be computed in an approximately constant time. By applying the node utility to the community formation game, we detect communities in large-scale networks efficiently. We also present a structural entropy-based node overlap heuristic function to detect overlapping communities, which can leverage the intermediate results of the community formation game to speed up the algorithm. Moreover, the proposed algorithms can apply to various graphs, whether unweighted, weighted, undirected, or directed graphs, to produce stable, reliable community structures in a unified framework. To our knowledge, CoDeSEG is the fastest known algorithm for large-scale network community detection. The algorithm's simplicity also supports straightforward parallelization, further enhancing its efficiency by computing the node strategies concurrently. Experiments conducted on several real-world networks show that our proposed algorithm consistently outperforms baselines in terms of performance. Moreover, the time overhead of CoDeSEG is significantly lower than that of the second-fastest baseline algorithm. The codes of CoDeSEG and baselines, along with datasets, are publicly available on GitHub[1]. In summary, the contributions of this paper are as follows:

• We propose a novel heuristic algorithm for community detection in large-scale networks, termed CoDeSEG. This algorithm introduces two-dimensional structural entropy to define the potential function of the community formation game and derives a node utility function with nearly constant time complexity.

• We design an efficient and effective two-stage algorithm for detecting overlapping communities in diverse graphs. Our algorithm identifies non-overlapping communities through the proposed community formation game and subsequently detects overlapping communities rapidly using a node overlap heuristic function based on structural entropy.

---

[1]https://anonymous.4open.science/r/CoDeSEG-6B06

• Experimental results on publicly available large-scale real-world networks demonstrate that the CoDeSEG algorithm outperforms state-of-the-art community detection algorithms regarding overlapping NMI and F1 scores, significantly reducing detection time. Compared to the fastest baseline method, CoDeSEG achieves an average speedup of 45 times in detection time.

## 2 PRELIMINARIES

In this section, we summarize the concepts related to the background of our work, including community detection, community formation games, and structural entropy. We summarize the glossary notations in Appendix A.

### 2.1 Community Detection

The goal of community detection is to identify communities such that the density of intra-community edges is higher than the density of inter-community edges, even when nodes belong to multiple communities. Given a graph $G = (\mathcal{V}, \mathcal{E})$, where $\mathcal{V}$ is the set of nodes (vertices), $\mathcal{E}$ is the set of edges (links) connecting the nodes, community detection algorithms find a set of communities $\mathcal{P} = \{\mathcal{C}_1, \mathcal{C}_2, \ldots, \mathcal{C}_k\}$, where each $\mathcal{C}_i \subseteq \mathcal{V}$ is a network community. In an overlapping community detection task, nodes $x \in \mathcal{V}$ can belong to more than one community.

### 2.2 Community Formation Game

Chen et al. [9] propose a game-theoretic-based community detection framework, named **community formation game**, that simulates the strategy selection and interactions of nodes within a network to identify community structures. In the game, each node $x \in \mathcal{V}$ is treated as a rational participant (player), consistently choosing the best strategy (community) that maximizes utility function. When the game converges to a Nash equilibrium, it corresponds to the communities the algorithm detects. We present relevant definitions, as follows:

*Definition 2.1 (Strategy Profile).* A strategy profile is a combination of strategies chosen by all players in the game. If there are $n$ players in the game, and each player $i$ has a set of strategies $S_i$, then a strategy profile $s$ is a tuple $\boldsymbol{s} = (s_1, s_2, \ldots, s_n)$, where $s_i \in S_i$ is the strategy chosen by player $i$.

*Definition 2.2 (Utility Function).* A utility (payoff) function represents the benefit a player receives based on the chosen strategies. For a player $i$, the utility function is denoted by $u_i : S \rightarrow \mathbb{R}$, where $S$ is the set of all possible strategy profiles. The function $u_i(s)$ gives the payoff to player $i$ when the strategy profile $\boldsymbol{s}$ is played.

*Definition 2.3 (Potential Game).* There exists a potential function $\varphi : S \rightarrow \mathbb{R}$, for any player $i$ and any two strategy profiles $\boldsymbol{s}$ and $\boldsymbol{s}'$ differing only in the strategy of player $i$, the change in the potential function equals the change of player $i$'s payoff:

$$\varphi(\boldsymbol{s}') - \varphi(\boldsymbol{s}) = u_i(\boldsymbol{s}') - u_i(\boldsymbol{s}) \tag{1}$$

where $u_i$ is the utility function for player $i$. Algorithms for learning in potential games, such as best response dynamics, can converge to a Nash equilibrium state, corresponding to the communities the algorithm identifies.

## 2.3 Structural Entropy

Structural entropy (SE) quantifies uncertainty and information content in complex networks, with lower values indicating more ordered structures and higher values reflecting greater disorder [26]. SE is defined on an encoding tree, where the encoding tree $\mathcal{T}$ of a graph $G = (\mathcal{V}, \mathcal{E})$ represents a hierarchical partition of $G$ and satisfies the following conditions:

(1) Each node $\alpha$ in $\mathcal{T}$ corresponds to a subset of nodes $T_\alpha \subseteq \mathcal{V}$. The root node $\lambda$ of $\mathcal{T}$ contains the entire set of nodes, i.e., $T_\lambda = \mathcal{V}$. Each leaf node $\gamma$ in $\mathcal{T}$ is associated with exactly one node from the graph $G$, meaning $T_\gamma = \{x\}$, where $x \in \mathcal{V}$.
(2) For each node $\alpha$ in $\mathcal{T}$, denote all its children as $\beta_1, \ldots, \beta_k$, then $T_{\beta_1}, \ldots, T_{\beta_k}$ is a partition of $T_\alpha$.
(3) For each node $\alpha$ in $\mathcal{T}$, denote its height as $h(\alpha)$. Let $h(\gamma) = 0$ and $h(\bar{\alpha}) = h(\alpha) + 1$, where $\bar{\alpha}$ is the parent of $\alpha$. The height of $\mathcal{T}$, $h(\mathcal{T}) = \max_{\alpha \in \mathcal{T}} h(\alpha)$.

The structural entropy (SE) of graph $G$ on encoding tree $\mathcal{T}$ is defined as:

$$\mathcal{H}^{\mathcal{T}}(G) = -\sum_{\alpha \in \mathcal{T}, \alpha \neq \lambda} \frac{g_\alpha}{v_\lambda} \log \frac{v_\alpha}{v_{\bar{\alpha}}}, \tag{2}$$

where $g_\alpha$ is the summation of the degrees (or weights) of the cut edges of $T_\alpha$ (edges that have exactly one endpoint in $T_\alpha$). $v_\alpha$, $v_{\bar{\alpha}}$, and $v_\lambda$ refer to the volumes of $T_\alpha$, $T_{\bar{\alpha}}$, and $T_\lambda$, respectively.

The $d$-dimensional structural entropy of $G$,

$$\mathcal{H}^{(d)}(G) = \min_{\forall \mathcal{T}: h(\mathcal{T}) = d} \{\mathcal{H}^{\mathcal{T}}(G)\}, \tag{3}$$

is realized by acquiring an optimal encoding tree of height $d$, in which the disturbance derived from noise or stochastic variation is minimized.

Communities within a network can be identified by minimizing its two-dimensional structural entropy. Suppose $\mathcal{P} = \{\mathcal{C}_1, \mathcal{C}_2, \ldots, \mathcal{C}_k\}$ is a partition of the network $G$, then the 2D structural entropy of $G$ is:

$$\mathcal{H}^2(\mathcal{P}) = -\sum_{c \in \mathcal{P}} \left( \frac{g_c}{v_\lambda} \log \frac{v_c}{v_\lambda} + \sum_{x \in c} \frac{d_x}{v_\lambda} \log \frac{d_x}{v_c} \right), \tag{4}$$

where $d_x$ is the degree of node $x$, $v_\lambda$ is the volume of the network.

## 3 METHODOLOGY

This section introduces the proposed algorithm CoDeSEG. Section 3.1 presents the structural entropy-based heuristic for the community formation game, followed by Section 3.2, which details key strategy computations. The full community detection algorithm is outlined in Section 3.3, and Section 3.4 analyzes its time complexity.

## 3.1 Structural Entropy based Heuristic Function

The proposed algorithm models community formation as a potential game, where the potential function is the network's 2D structural entropy $\mathcal{H}^2(\mathcal{P})$. Each node selects the community that most reduces this entropy as its optimal strategy. When the game converges to a Nash equilibrium, yielding communities with a minimized two-dimensional structural entropy.

Consider a node in $G$ adopting a strategy, such as altering its community membership, resulting in a new partition denoted by $\mathcal{P}'$.

We define the **heuristic function** $\Delta$ as the change in the potential function.

$$\Delta = \mathcal{H}^2(\mathcal{P}) - \mathcal{H}^2(\mathcal{P}'). \tag{5}$$

In the disjoint community formation game, each node aims to maximize the value of $\Delta$ by moving to the best adjacent community, resulting in a partition with reduced 2D structural entropy. A node can choose from three strategies: **Stay**, **Leave and be alone**, and **Transfer to another community**.

**Stay**: Node $x$ decides to stay in the current community, then the partition $\mathcal{P}$ remains unchanged, $\mathcal{P}' = \mathcal{P}$. The value of heuristic function $\Delta_S$ is:

$$\Delta_S = \mathcal{H}^2(\mathcal{P}) - \mathcal{H}^2(\mathcal{P}') = 0. \tag{6}$$

**Leave and be alone**: Suppose the original partition is $\mathcal{P} = \{\mathcal{C}_1, \mathcal{C}_2, \ldots, \mathcal{C}_k\}$ and when node $x$ leaves its community $\mathcal{C}_k$, resulting a new partition $\mathcal{P}' = \{\mathcal{C}_1, \mathcal{C}_2, \ldots, \mathcal{C}'_k, \{x\}\}$, where $\mathcal{C}_k = \mathcal{C}'_k \cup \{x\}$. The value of heuristic function $\Delta_L(x, \mathcal{C}_k)$ is:

$$\begin{aligned} \Delta_L(x, \mathcal{C}_k) =& \mathcal{H}^2(\mathcal{P}) - \mathcal{H}^2(\mathcal{P}') \\ =& \mathcal{H}^2(\mathcal{C}_k) - \mathcal{H}^2(\mathcal{C}'_k) - \mathcal{H}^2(\{x\}). \end{aligned} \tag{7}$$

The calculation details for Equation (7) are provided in Section 3.2. If community $\mathcal{C}_k$ is a singleton, then $\Delta_L(x, \mathcal{C}_k) = 0$.

**Transfer to another community:** Suppose $x$ transfers from $\mathcal{C}_1$ to $\mathcal{C}_k$, the original partition is $\mathcal{P} = \{\mathcal{C}_1, \mathcal{C}_2, \ldots, \mathcal{C}_k\}$ and the new partition is $\mathcal{P}' = \{\mathcal{C}'_1, \mathcal{C}_2, \ldots, \mathcal{C}'_k\}$, where $\mathcal{C}'_1 = \mathcal{C}_1 \backslash \{x\}$, $\mathcal{C}'_k = \mathcal{C}_k \cup \{x\}$. The transfer strategy can be seen as a composite transformation of two steps. Firstly, node $x$ leaves $\mathcal{C}_1$ and does not join any community, which yields an intermediate $\mathcal{P}^m = \{\mathcal{C}'_1, \mathcal{C}_2, \ldots, \mathcal{C}_k, \{x\}\}$. Secondly, node $x$ join $\mathcal{C}_k$, resulting in new partition $\mathcal{P}$. We can easily figure out that the second step is an inverse transformation of the Leave and be alone strategy. Therefore, the value of the heuristic function $\Delta_T(x, \mathcal{C}_1, \mathcal{C}_k)$ can be expressed as:

$$\begin{aligned} \Delta_T(x, \mathcal{C}_1, \mathcal{C}_k) &= \mathcal{H}^2(\mathcal{P}) - \mathcal{H}^2(\mathcal{P}') \\ &= (\mathcal{H}^2(\mathcal{P}) - \mathcal{H}^2(\mathcal{P}^m)) + (\mathcal{H}^2(\mathcal{P}^m) - \mathcal{H}^2(\mathcal{P}')) \\ &= \Delta_L(x, \mathcal{C}_1) - \Delta_L(x, \mathcal{C}_k). \end{aligned} \tag{8}$$

Since a node $x$ may have several adjective target communities, we denote the best-transferring with max $\Delta_T$ as $\Delta_T(x, \mathcal{C}_{\text{best}})$.

In our community formation game, a node will only join a new community if it decreases the network's 2D structural entropy. Consequently, a node will prefer to stay in its current community unless another community offers a further reduction in 2D structural entropy. Thus, Leave and be alone strategy is optional. Therefore, in our disjoint community formation game algorithm, node $x$ selects its movement strategy according to the following formula:

$$S(x) = \max(\Delta_S, \Delta_T(x, \mathcal{C}_{\text{best}})). \tag{9}$$

After identifying disjoint communities, we propose using a structural entropy heuristic to determine whether a node $x$ should overlap between communities.

**Overlap nodes:** Suppose the partition is $\mathcal{P} = \{\mathcal{C}_1, \mathcal{C}_2, \ldots, \mathcal{C}_k\}$ and a node $x \notin \mathcal{C}_k$. If we copy $x$ to community $\mathcal{C}_k$, we create a new overlapping partition $\mathcal{P}' = \{\mathcal{C}_1, \mathcal{C}_2, \ldots, \mathcal{C}'_k\}$, where $\mathcal{C}'_k = \mathcal{C}_k \cup \{x\}$. Since the copy action does not affect the origin community, we define the overlapping heuristic function as follows:

$$\Delta_O(x, \mathcal{C}_k) = \mathcal{H}^2(\mathcal{P}) - \mathcal{H}^2(\mathcal{P}') = -\Delta_L(x, \mathcal{C}'_k). \tag{10}$$

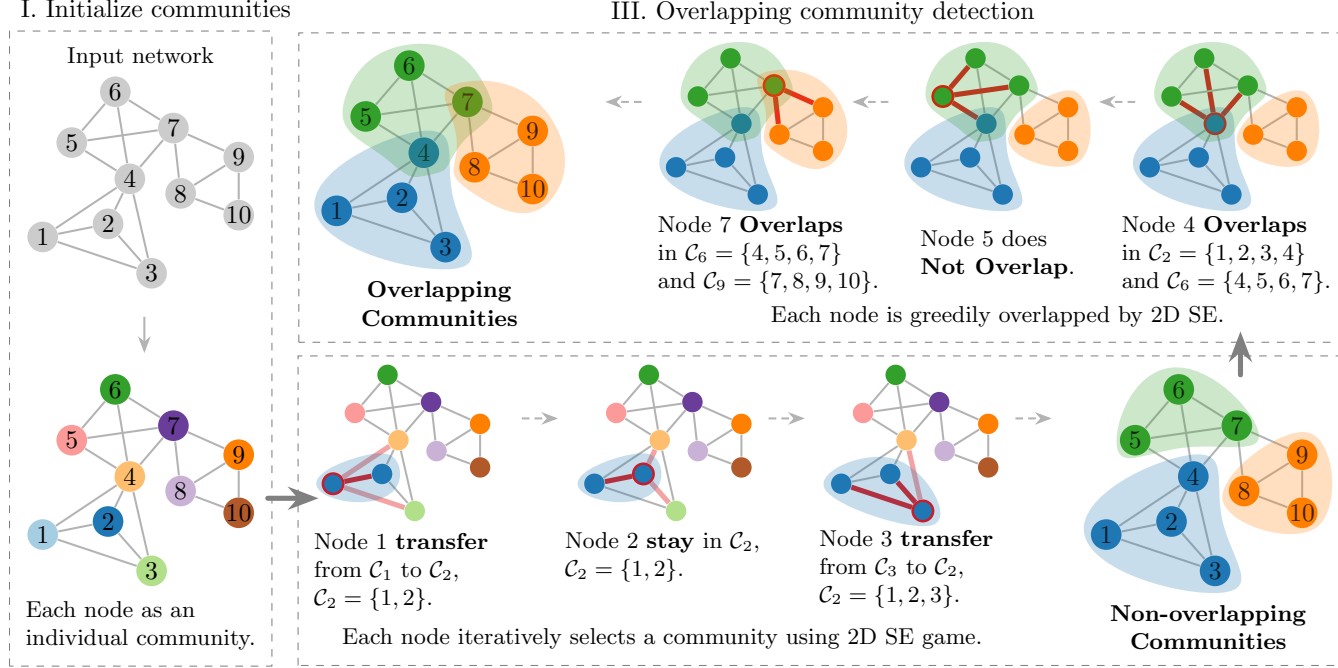

**Figure 2: Overview of the proposed CoDeSEG algorithm.**

If $\Delta_O(x, C_k) > 0$, it means the overlap action reduces the partition's structural entropy. However, this criterion may allow excessive node overlapping. To address this, we propose using the average value of $\Delta_L(x_i, C_k)$ for nodes in the community $C_k$ as a threshold. Nodes can only overlap with $C_k$ if their $\Delta_L(x, C_k)$ exceeds this threshold.

### 3.2 Efficient computation of $\Delta_L(x, C_k)$

The formulas for the strategies above highlight that efficiently completing the community formation game depends on quickly computing the node leave strategy. By deriving Equation (7), we obtain an efficient formula for computing $\Delta_L(x, C_k)$:

$$
\begin{aligned}
\Delta_L(x, C_k) &= \mathcal{H}^2(\mathcal{P}) - \mathcal{H}^2(\mathcal{P}') \\
&= \mathcal{H}^2(C_k) - \mathcal{H}^2(C_k') - \mathcal{H}^2(\{x\}) \\
&= \frac{g_{c_k'}}{v_\lambda} \log \frac{v_{c_k'}}{v_\lambda} - \frac{g_{c_k}}{v_\lambda} \log \frac{v_{c_k}}{v_\lambda} + \frac{d_x}{v_\lambda} \log \frac{v_{c_k}}{v_\lambda} + \frac{v_{c_k'}}{v_\lambda} \log \frac{v_{c_k}}{v_{c_k'}},
\end{aligned}
\tag{11}
$$

where $v_{c_k'}$ represents the volume of $C_k'$, and $g_{c_k'}$ denotes the sum of the degrees (weights) of the cut edges of $C_k'$. The detailed derivation is provided in Appendix B.

By caching all community volumes $\{v_{c_1}, v_{c_2}, \ldots, v_{c_k}\}$ and cut edge summations $\{g_{c_1}, g_{c_2}, \ldots, g_{c_k}\}$, computing $v_{c_k'}$ and $g_{c_k'}$ becomes straightforward, allowing us to calculate $\Delta_L(x, C_k)$ in constant time complexity, $O(1)$. For an undirected graph, $v_{c_k'}$ and $g_{c_k'}$ can be computed using the following equations:

$$
v_{c_k'} = v_{c_k} - d_x, \quad g_{c_k'} = g_{c_k} + 2d_x^{in} - d_x,
\tag{12}
$$

where $d_x^{in}$ denotes the sum of edge weights between node $x$ and its neighbor nodes within community $C_k$.

The proposed strategies can be easily adapted for directed networks by separately considering the in-degree and out-degree of node $x$ relative to community $C_k$. Consequently, Equation (12) is updated accordingly:

$$
v_{c_k'} = v_{c_k} - d_x, \quad g_{c_k'} = g_{c_k} + d_x^{in} + d_x^{out} - d_x.
\tag{13}
$$

Once a node $x$ is transferred to a new community, we will update the statistics of the target and source communities using the following formulas,

$$
\begin{aligned}
v_{c_t} &\leftarrow v_{c_t} + d_x, & v_{c_k} &\leftarrow v_{c_k'}, \\
g_{c_t} &\leftarrow g_{c_t} - 2d_x^{in} + d_x, & g_{c_k} &\leftarrow g_{c_k'}.
\end{aligned}
\tag{14}
$$

For directed graphs, we can easily derive similar formulas for updating community statistics.

### 3.3 Community Detection Algorithm

After developing effective methodologies for community formation games, we introduce a new two-stage algorithm for overlapping community detection. Figure 2 provides an overview of this algorithm. In the non-overlapping detection phase, each node $x$ sequentially implements the best strategy from Equation (9) until all nodes are assigned to their communities. In the overlapping phase, nodes $x$ can overlap multiple communities if the overlap action meets the specified threshold.

*3.3.1 Non-overlapping Community Detection.* We propose a non-overlapping community detection algorithm designed to minimize 2D structural entropy using a potential game, where the optimal

strategy is computed by Equation (9). Once the game reaches a Nash equilibrium, the communities stabilize and no longer change. The pseudo-code of the proposed algorithm is provided in Algorithm 1.

In Algorithm 1, we initialize each node as an individual cluster and compute their volumes, setting the cut edges' summations as the node degrees (lines 1-3). In a directed network, the volumes of these communities are node in-degrees, and the summations of cut edges are node out-degrees.

The heart of our algorithm lies in an iterative loop of community formation games. We evaluate every node $x$ in each iteration and determine the optimal strategy for $x$ to significantly minimize the 2D structural entropy of the graph (lines 7-17). To get the best strategy for a node, we firstly compute the heuristic $\Delta_L(x, C_x)$ that provides a measure of the impact when node $x$ leaves its current community. The maximum heuristic function value $\Delta_{\max}$ is initially set to $\Delta_L(x, C_x)$, and the target community index $t$ is set to $t_c$ indicating stay in the current community (lines 7-11). Then, we evaluate each adjacent community $C_k$ to determine if moving node $x$ to any of these communities would result in a greater heuristic function value (lines 12-17). For each $C_k$, we compute the transfer heuristic $\Delta_T(x, C_x, C_k)$ and compare it to the maximum heuristic function value. If moving to an adjacent community $C_k$ offers a larger heuristic function value, we update $\Delta_{\max}$ and set $t$ to the

---

**Algorithm 1:** Non-overlapping Community Detection.

**Input:** Graph $G = (\mathcal{V}, \mathcal{E})$.
**Output:** Non-overlapping communities (partition) $\mathcal{P}$ of $G$.

1  $\mathcal{P} \leftarrow$ Each node as an individual community.
2  Initialize community volumes $\{v_{c_1}, v_{c_2}, \ldots, v_{c_n}\}$.
3  Initialize cut edge summations $\{g_{c_1}, g_{c_2}, \ldots, g_{c_n}\}$.
4  **while** *true* **do**
5  $\quad$ $\Delta_{\text{sum}} \leftarrow 0, M \leftarrow 0$
6  $\quad$ **for** *node $x \in \mathcal{V}$* **do**
7  $\quad\quad$ $C_x \leftarrow$ Community contains $x$
8  $\quad\quad$ $t_c \leftarrow$ Index of community $C_x$
9  $\quad\quad$ $t \leftarrow t_c$ $\qquad\qquad$ ▷ $t_c$ means stay in $C_x$
10 $\quad\quad$ $\Delta_L(x, C_x) \leftarrow$ Eq.11
11 $\quad\quad$ $\Delta_{\max} \leftarrow \Delta_L(x, C_x)$
12 $\quad\quad$ **for** *k-th adjacent community $C_k$ of node $x$* **do**
13 $\quad\quad\quad$ $\Delta_L(x, C_k) \leftarrow$ Eq.11
14 $\quad\quad\quad$ $\Delta_T(x, C_x, C_k) \leftarrow$ Eq.8
15 $\quad\quad\quad$ **if** $\Delta_T(x, C_x, C_k) > \Delta_{\max}$ **then**
16 $\quad\quad\quad\quad$ $\Delta_{\max} \leftarrow \Delta_T(x, C_x, C_k)$
17 $\quad\quad\quad\quad$ $t \leftarrow k$
18 $\quad\quad$ **if** $t \neq t_c$ **then**
19 $\quad\quad\quad$ Transfer $x$ from $C_x$ to $C_t$
20 $\quad\quad\quad$ Update statistics of $C_x, C_t$ by Eq.14
21 $\quad\quad\quad$ $M \leftarrow M + 1$.
22 $\quad\quad$ $\Delta_{\text{sum}} \leftarrow \Delta_{\text{sum}} + \Delta_{\max}$
23 $\quad$ **if** $M = 0$ or Eq. 15 is satisfied **then**
24 $\quad\quad$ Break
25 **return** $\mathcal{P}$

---

**Algorithm 2:** Overlapping community detection.

**Input:** Non-overlapping communities $\mathcal{P}$.
**Output:** Overlapping communities $\mathcal{P}^o$

1  $\mathcal{P}^o \leftarrow \mathcal{P}$
2  **for** *node $x$ in $\mathcal{V}$* **do**
3  $\quad$ **for** *k-th adjacent community $C_k$ of node $x$* **do**
4  $\quad\quad$ $\Delta_O(x, C_k) \leftarrow$ Eq.10
5  $\quad\quad$ **if** $\Delta_O(x, C_k) > \tau_o$ **then**
6  $\quad\quad\quad$ Overlap node $x$ to $C_k$ in $\mathcal{P}^o$
7  **return** $\mathcal{P}^o$

---

index of community $C_k$ (lines 15-17). Finally, we find the optima index $t$ and the maximum heuristic function value $\Delta_{\max}$.

Suppose the strategy indicates that node $x$ should move to another community. We transfer $x$ from its current community $C_x$ to the target community $C_t$ (line 19) and update the statistics of the source and target communities, such as adjusting community volumes and cut edge summations via Equation (14) (line 20).

At the end of each iteration, if all nodes choose to stay in their current community (i.e., $M = 0$), the algorithm is considered converged. In practice, we introduce a stopping criterion: if the average of $\Delta_T$ decreases significantly compared to the initial average node entropy, it suggests that further adjustments will have little impact on improving the community structure. Formally, the stopping criterion is defined as:

$$\frac{\Delta_{sum}}{M} \leq \frac{\tau_n}{|V|} \sum_{x \in V} -\frac{d_x}{v_\lambda} \cdot \log \frac{d_x}{v_\lambda}, \tag{15}$$

where $\Delta_{sum}$ represents the change in the graph's 2D structural entropy during the current iteration, $M$ denotes the number of nodes that changed communities, and $\tau_n$ is a hyper-parameter, with a range of $(0, 1)$. In the end, our algorithm outputs the partition $\mathcal{P}$, representing the final assignment of nodes into non-overlapping communities.

*3.3.2 Overlapping Community Detection.* The overlapping community detection algorithm begins with a non-overlapping partition $\mathcal{P}$ of the network $G$. The goal of the overlapping community detection (Algorithm 2) is to generate a set of overlapping communities $\mathcal{P}^o$ of $G$. Initially, $\mathcal{P}^o$ is identical to $\mathcal{P}$. For each node $x$ in the network, the algorithm iterates over all its adjacent communities $C_k$. It computes the overlapping heuristic function $\Delta_O(x, C_k)$. If the heuristic function exceeds the overlap threshold $\tau_o$, we overlap node $x$ to community $C_k$. We define the overlap threshold $\tau_o$ as the average of node heuristic function values,

$$\tau_o = \frac{1}{|C_k|} \sum_{x_i \in C_k} -\Delta_L(x_i, C_k). \tag{16}$$

The iterative process ensures that nodes overlap in communities, significantly reducing graph 2D structural entropy.

## 3.4 Time Complexity

The time complexity of the proposed community detection algorithm is $O(I \cdot d_{\max} \cdot N)$, where $N$ denotes the number of nodes in the graph, $d_{\max}$ represents the maximum degree of any node, and $I$

is the number of iterations required for the algorithm to converge to a stable partition.

In the non-overlapping detection phase (Algorithm 1), the algorithm initially sets each node as an individual cluster and initializes community volumes and cut edge summations, which takes $O(N)$ time. For each node, the best strategy computation for each node depends on its degree, taking $O(d_{avg})$ time, since we can compute $\Delta_L(x, C)$ in $O(1)$ time (Section 3.2). Therefore, evaluating and updating all nodes in one iteration requires $O(d_{avg} \cdot N)$. Overall, given that the algorithm runs for $I$ iterations, the total time complexity is $O(I \cdot d_{avg} \cdot N)$. This complexity indicates that the algorithm's performance scales linearly with the number of nodes and their average degree, with the number of iterations needed for convergence influencing the overall computational effort. In the overlapping community detection phase, the algorithm's time complexity is $O(d_{avg} \cdot N)$. This complexity arises because each node is processed by iterating over its adjacent communities.

Owing to each node's greedy selection of the most suitable community, our algorithm converges quickly, typically in fewer than 10 iterations (see the experiment in section 4.4). Moreover, in large-scale real-world networks, $d_{avg} \ll N$, which reduces the algorithm's complexity to almost $O(N)$ for many networks. This allows our algorithm to detect overlapping communities on graphs with millions of nodes in just seconds.

## 3.5 Parallelism Implementation

Parallelization can fully exploit the multi-core architecture of modern CPUs, significantly reducing the algorithm's runtime. The proposed algorithm is readily parallelizable. In our parallelized CoDeSEG algorithm, each node in different threads independently calculates the optimal strategy based on the current partition. During the execution of the movement strategy, a mutex lock ensures the correct update of statistics such as $C_k$ and $g_c$. It is crucial to note that when the volume and cut of a community are concurrently updated by multiple threads, the increments must be recalculated. However, it is worth mentioning that as the algorithm progresses, the number of nodes requiring movement decreases sharply, thereby enhancing the acceleration effect of parallelism.

During the overlapping community detection phase, our algorithm is inherently parallelizable. A node's inclusion in an overlapping community is determined by the change in entropy resulting from its addition, which depends only on the outcomes of non-overlapping community detection and the node's connections to those communities. This allows the overlapping strategies for each node to be calculated concurrently. Experiments in section 4.5 show that parallel CoDeSEG can significantly reduce the run time on large networks.

## 4 EXPERIMENTS

In this section, we conduct extensive experiments to validate the effectiveness and superiority of the proposed algorithm. Our goal is to address the following four key research questions: **RQ1**: How does the CoDeSEG perform in overlapping and non-overlapping community detection tasks compared to baselines? **RQ2**: How does the detection efficiency of CoDeSEG on different networks compare to the baselines? **RQ3**: Can CoDeSEG achieve fast convergence,

**Table 1: Statistics of datasets.**

| Dataset | #Nodes | #Edges | Avg. Deg | #Cmty |
|---|---|---|---|---|
| Amazon | 334,863 | 925,872 | 5.530 | 75,149 |
| YouTube | 1,134,890 | 2,987,624 | 5.265 | 8,385 |
| DBLP | 317,080 | 1,049,866 | 6.622 | 13,477 |
| LiveJournal | 3,997,962 | 34,681,189 | 17.349 | 287,512 |
| Orkut | 3,072,441 | 117,185,083 | 76.281 | 6,288,363 |
| Friendster | 65,608,366 | 1,806,067,135 | 55.056 | 957,154 |
| Wiki | 1,791,489 | 28,511,807 | 15.915 | 17,364 |
| Tweet12 | 68,841 | 10,141,672 | 147.320 | 503 |
| Tweet18 | 64,516 | 17,926,784 | 277.866 | 257 |

and how do key hyperparameters impact its performance? **RQ4**: How does parallelization of CoDeSEG influence its performance and efficiency?

### 4.1 Experimental Setups

*4.1.1 Evaluation Metrics.* For datasets with ground truth, the main goal of the experiment is to evaluate how closely the detected results match the ground truth. We use two evaluation metrics: the Average F1-Score (F1) [30, 50] and Overlapping Normalized Mutual Information (ONMI) [24]. For non-overlapping detection tasks, we use the non-overlapping NMI. More details on these metrics are provided in Appendix C.

*4.1.2 Datasets.* We conduct overlapping community detection experiments on seven large-scale unweighted networks with ground truth from SNAP [21], where the Wiki dataset is a directed network, and the others are undirected and unweighted. Additionally, we perform non-overlapping community detection experiments on two real-world weighted social networks: Tweet12 [36] and Tweet18 [34]. The dataset statistics are shown in Table 1, with detailed descriptions provided in Appendix D.

*4.1.3 Baselines.* To assess the performance of the proposed algorithm across various networks, we compare it with four sophisticated overlapping community detection methods (**SLPA** [49], **Bigclam** [50], **NcGame** [13], and **Fox** [31]) and four proven non-overlapping community detection methods (**Louvain** [7], **DER** [23], **Leiden** [43], and **FLPA** [44]). For additional experimental details, including the hyperparameter settings and descriptions of the baseline algorithms, please refer to Appendix E.

*4.1.4 Implementation.* All experiments are conducted on a server with a hardware configuration of dual 16-core Intel Xeon Silver 4314 processors @ 2.40GHz and 1024GB of memory. For CoDeSEG, we set the termination threshold $\tau_n$ to 0.3.

### 4.2 Main Results (RQ1)

Tables 2 and 3 illustrate that CoDeSEG performs exceptionally in overlapping and non-overlapping community detection tasks. In the overlapping detection scenario, it consistently ranks first or second in ONMI and F1 scores across all seven datasets, particularly excelling in YouTube, Orkut, Friendster, and Wiki, where it achieves the highest values. Compared to label propagation-based

**Table 2: Results on unweighted overlapping networks (%). The best results are bolded, and the second-best results are underlined. \* indicates the results when treating directed networks as undirected. N/A indicates the runtime extended beyond a week.**

| Dataset | Amazon | | YouTube | | DBLP | | LiveJournal | | Orkut | | Friendster | | Wiki | |
|---------|--------|------|---------|------|------|------|-------------|------|-------|------|------------|------|------|------|
| Metric | ONMI | F1 | ONMI | F1 | ONMI | F1 | ONMI | F1 | ONMI | F1 | ONMI | F1 | ONMI | F1 |
| SLPA | 9.05 | 34.37 | 4.58 | 26.16 | 4.45 | 23.27 | 2.55 | 22.29 | 0.12 | 8.51 | 0.03 | 2.34 | 0.48* | 12.16* |
| Bigclam | 7.62 | 33.30 | 1.47 | 27.78 | 5.96 | 24.91 | 2.05 | 10.67 | 0.10 | 11.72 | N/A | N/A | N/A | N/A |
| NcGame | 9.94 | **36.01** | 1.23 | 11.37 | 4.89 | 23.94 | 1.33 | 7.96 | 0.07 | 2.20 | 0.19 | 0.10 | 0.02* | 10.96* |
| Fox | 8.88 | 29.04 | 6.67 | 31.77 | **7.34** | 23.85 | **4.18** | 26.79 | 0.47 | 24.07 | 0.56 | 19.03 | 0.45* | 10.80* |
| CoDeSEG | **10.75** | 34.51 | **8.17** | **36.80** | 7.21 | **25.34** | 3.75 | **28.46** | **0.49** | **25.26** | **0.73** | **19.21** | **2.28** | **18.92** |
| Improve | ↑ 0.81 | ↓ 1.50 | ↑ 1.50 | ↑ 5.03 | ↓ 0.13 | ↑ 1.23 | ↓ 0.43 | ↑ 1.67 | ↑ 0.02 | ↑ 1.19 | ↑ 0.17 | ↑ 0.18 | ↑ 1.83 | ↑ 6.76 |

**Table 3: Results on weighted non-overlapping networks (%).**

| Method | | Louvain | DER | Leiden | FLPA | CoDeSEG | Improve |
|--------|-----|---------|-------|--------|-------|---------|---------|
| Twe-et12 | NMI | 52.93 | 10.62 | 54.37 | 79.39 | **83.41** | ↑ 4.02 |
| | F1 | 39.63 | 19.20 | 42.01 | 65.14 | **66.61** | ↑ 1.47 |
| Twe-et18 | NMI | 47.85 | 11.94 | 48.81 | 49.25 | **73.27** | ↑ 24.02 |
| | F1 | 53.60 | 32.76 | 52.13 | 65.05 | **69.77** | ↑ 4.72 |

SLPA and Non-negative Matrix Factorization-based Bigclam, CoDeSEG consistently outperforms these algorithms in ONMI and F1 scores. Notably, SLPA and Bigclam experience significant performance declines on larger networks such as Orkut and Friendster. While NcGame records a higher F1 score on the Amazon dataset, its performance is inconsistent across other datasets. Additionally, FOX, which ranks second to CoDeSEG, relies on a complex heuristic that impedes efficiency. CoDeSEG uniquely supports community detection in the large-scale directed network Wiki, demonstrating a significant performance drop in baseline methods when Wiki is treated as an undirected network. Specifically, CoDeSEG enhances ONMI by 381.25% and F1 by 55.59% compared to the best baseline, SLPA, highlighting its advantages in directed network contexts.

In the non-overlapping detection scenario (as shown in Table 3), CoDeSEG outperforms all baseline algorithms, showcasing superior stability and robustness. Compared to the second-best method, FLPA, CoDeSEG achieves average improvements of 21.80% and 4.75% in NMI and F1 scores, respectively. Compared to modularity-based methods (Leiden and Louvain), it shows even greater improvements, with average gains of 51.85% to 69.27% in NMI and 44.86% to 46.28% in F1 scores. Moreover, DER's performance on both the Tweet12 and Tweet18 networks is notably inferior to that of CoDeGSE. Overall, CoDeSEG's superior performance underscores the effectiveness of the utility function that combines potential games with structural entropy in uncovering network community structures. Further details on network visual analysis can be found in Appendix F.

## 4.3 Efficiency of CoDeSEG (RQ2)

As shown in Table 4, the detection efficiency of CoDeGSE significantly surpasses that of all baseline methods, with efficiency gains increasing as the network size expands. In the overlapping network detection tasks, CoDeGSE exhibits particularly substantial efficiency improvements on networks such as LiveJournal, Orkut, Friendster, and Wiki. On average, CoDeSEG is 45 times faster than

**Table 4: Time consumption of different algorithms (Sec).**

| Unweighted overlapping networks | | | | | | |
|---------|---------|--------|--------|--------|---------|-------|
| Dataset | Bigclam | SLPA | NcGame | Fox | CoDeSEG | Ratio |
| Amazon | 482 | 369 | 44 | 15 | **2** | 7.5 |
| YouTube | 442764 | 852 | 1374 | 220 | **6** | 36.7 |
| DBLP | 2106 | 263 | 62 | 15 | **2** | 7.5 |
| LiveJournal | 1064 | 11343 | 4147 | 2330 | **51** | 20.9 |
| Orkut | 2899 | 21429 | 37590 | 73385 | **279** | 10.4 |
| Friendster | > 7day | 298536 | 191556 | 487397 | **5650** | 33.9 |
| Wiki | > 7day | 4410 | 36233 | 15511 | **27** | 163.3 |

| Weighted non-overlapping networks | | | | | | |
|---------|---------|------|--------|------|---------|-------|
| Dataset | Louvain | DER | Leiden | FLPA | CoDeSEG | Ratio |
| Tweet12 | 26 | 3675 | 66 | 20 | **12** | 1.7 |
| Tweet18 | 47 | 7622 | 149 | 45 | **15** | 3.0 |

the quickest baseline, NcGame. The proposed CoDeSEG converges rapidly in just a few iterations, while Bigclam requires many more. Moreover, due to the uncertainty in Bigclam's convergence process, its detection time on the YouTube network is considerably longer than on larger networks like LiveJournal and Orkut. For SLPA, the detection speed is constrained by the predefined number of iterations. NcGame fails to dynamically update necessary variables during detection, resulting in additional computation time on large networks. Additionally, for Fox, detection time increases proportionately to the number of overlapping nodes.

In detecting non-overlapping networks, CoDeGSE achieves a 418-fold improvement in efficiency compared to DER, while its efficiency increases by 2.4 times compared to the fastest baseline, FLPA. The relatively modest improvement in efficiency compared to FLPA can be attributed to the smaller scales of the Tweet12 and Tweet18 networks, where the efficient derivation of the CoDeGSE utility function demonstrates greater advantages in larger-scale networks. CoDeSEG's superior performance and efficiency make it highly scalable for large, complex, real-world networks.

## 4.4 Convergence of CoDeSEG (RQ3)

Figure 3 presents the convergence of CoDeSEG during the non-overlapping detection phase on the Amazon and DBLP networks. The structural entropy and node movements drop sharply within the first three iterations, with convergence achieved by the fifth

iteration, where moved nodes represent only 1/200 of the total. The algorithm's linear complexity and rapid convergence make it highly efficient for large-scale complex networks. Furthermore, CoDe-SEG's stable convergence process has no node repeated adjustments in the last iteration and yields an accurate Nash equilibrium.

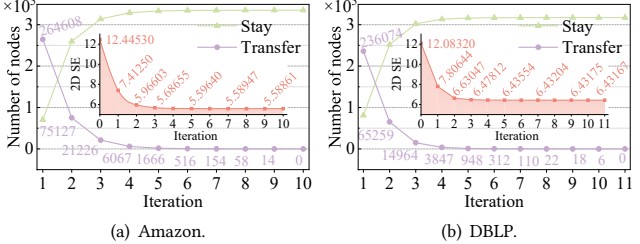

Figure 3: Convergence of CoDeSEG on Amazon and DBLP.

The influence of a few unstable nodes during the non-overlapping detection phase is minimal. Although some nodes may not join optimal communities initially, they are correctly replicated during the overlapping phase. As Figure 4 shows, the F1 score stabilizes by the 5th iteration for Amazon and the 4th for DBLP. We also analyze the termination threshold $\tau_n$, finding that a high $\tau_n$ (e.g., 0.4) may cause premature termination and reduced performance, while a low $\tau_n$ (e.g., 0.2) increases iteration time. A balanced $\tau_n$ (e.g., 0.3) can maintain performance while improving detection efficiency.

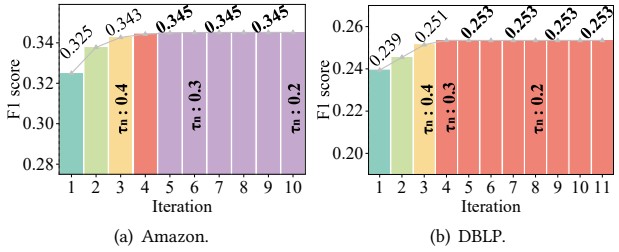

Figure 4: F1 score of each iteration on Amazon and DBLP.

## 4.5 Parallelization Study (RQ4)

The primary objective of parallelizing the algorithm is to significantly reduce runtime while preserving as much of the algorithm's performance as possible. To assess the impact of varying thread counts on performance and efficiency, we conduct experiments on the LiveJournal and Orkut networks using different thread configurations (1, 2, 4, 8, 16, 32, 64), as presented in Figure 5. The results reveal that although CoDeSEG's partitioning outcomes show minor variations across different thread counts, these differences are negligible compared to the ground truth communities. Additionally, runtime reduction does not follow a strictly linear pattern as thread count increases. For instance, runtime with 32 threads is shorter in both networks than 64 threads. This can be attributed to the fact that as thread count rises, computational discrepancies between threads may lead to an increase in the number of iterations, thus prolonging the total runtime. Moreover, with more threads, overhead related to thread initialization, lock contention, and synchronization delays also rise. Therefore, balancing task partitioning granularity and the efficiency gains of parallelization is crucial when determining the optimal number of threads.

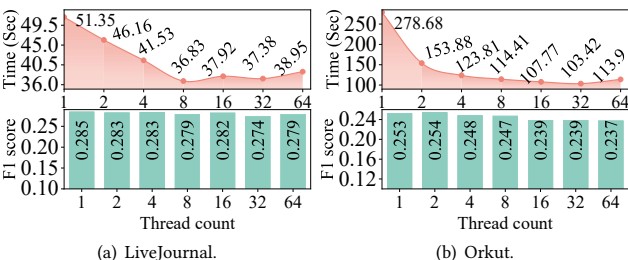

Figure 5: F1 scores and runtime on LiveJournal and Orkut with different threads.

## 5 RELATED WORK

Community detection has a 20-year history, during which numerous algorithms have been developed using various methods such as modularity [7, 10, 18, 43], label propagation [28, 44, 49], seed expansion [2, 19, 47, 51], non-negative matrix factorization [6, 29, 50], spectral clustering [5, 27, 46]. However, community detection in large-scale networks remains a challenging task.

Game theory-based community detection methods focus on decision-making processes where one agent's choice affects others. Early approaches like Chen's non-cooperative game-based algorithm inspired further work on utility functions for disjoint community detection [9]. Alvari et al. [1] propose to use structural equivalence to detect overlapping community structures, while Crampes et al. [11] introduce a potential game for node reassignment, although it requires knowing the number of clusters. However, most game-theoretic-based algorithms proposed in the last decay are not scalable for large networks.

For large-scale community detection, Lyu et al. propose the FOX [31] algorithm, which measures the closeness between nodes and communities by approximating the number of triangles in communities. Ferdowsi et al. [13] propose a two-phase non-cooperative game model that detects non-overlapping communities by a local interaction utility function, then identifies overlapping nodes leverage payoffs acquired from communities membership. In contrast to existing approaches, we propose a utility function based on structural entropy that facilitates efficient community detection, while accounting for the global partition of the network.

## 6 CONCLUSION

In this paper, we propose a fast heuristic community detection algorithm by minimizing the two-dimensional structural entropy of networks within the framework of a potential game. By designing a utility function with nearly linear time complexity, our algorithm can efficiently detect high-quality communities in large-scale networks, completing the process within minutes, even for networks comprising millions of nodes. Experimental results on real-world datasets demonstrate its practicality and efficiency. We envision broad applications for the proposed algorithm across diverse fields, including social networks, biomedicine, and e-commerce. While this study focuses on community detection in static graphs, an important future direction would be to extend the algorithmic framework to dynamic graph community detection, further enhancing its utility in evolving network structures.

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

# A GLOSSARY OF NOTATIONS

Notations used in this paper, along with their corresponding description, are presented in Table 5.

**Table 5: Glossary of Notations.**

| Notation | Description |
|---|---|
| $G$ | Network or graph |
| $\mathcal{V}$ | Nodes (vertices) in network $G$ |
| $\mathcal{E}$ | Edges (links) in network $G$ |
| $\mathcal{P}$ | A set of communities of network $G$ |
| $C_i$ | The $i$-th community in $\mathcal{P}$ |
| $S_i$ | The strategy of the node (player) $i$ |
| $s$ | A strategy profile combines the strategies chosen by all the players in the game |
| $u_i$ | The payoff function of player $i$ |
| $\varphi$ | A potential function in the potential game |
| $\mathcal{T}$ | An encoding tree of the graph $G$ |
| $\alpha$ | A node in the encoding tree $\mathcal{T}$ |
| $\bar{\alpha}$ | The parent node of the node $\alpha$ in the encoding tree |
| $\lambda$ | The root node of the encoding tree $\mathcal{T}$ |
| $T_\alpha$ | A set of vertices in the encoding tree node $\alpha$ |
| $h(\alpha)$ | The height of the tree node $\alpha$ |
| $\mathcal{H}(G)$ | The structural entropy (SE) of graph $G$ on the encoding tree |
| $\mathcal{H}^{(d)}(G)$ | The $d$-dimensional structural entropy of $G$ |
| $\mathcal{H}^2(\mathcal{P})$ | The 2D structural entropy of the graph partition $\mathcal{P}$ |
| $g_\alpha$ | The summation of the degrees (weights) of the cut edges of $T_\alpha$ |
| $d_x$ | The degree of the node $x$ in the graph $G$ |
| $v_\alpha, v_{\bar{\alpha}}, v_\lambda$ | The summation of the degrees (weights) of encoding tree nodes, $\alpha$, $\bar{\alpha}$, and $\lambda$ |
| $\Delta_S, \Delta_L, \Delta_T$ | Heuristic functions for strategies: Stay, Leave and be alone, and Transfer to another community |
| $\Delta_O$ | Heuristic function for overlapping nodes |
| $\tau_n, \tau_o$ | the termination threshold, the overlap threshold |

# B PROOF OF THE $\Delta_L(x, \mathcal{C})$ COMPUTATION FORMULA

Suppose the original partition is $\mathcal{P} = \{\mathcal{C}_1, \mathcal{C}_2, \ldots, \mathcal{C}_k\}$ and when node $x$ leaves $\mathcal{C}_k$, forming a new partition $\mathcal{P}' = \{\mathcal{C}_1, \mathcal{C}_2, \ldots, \mathcal{C}'_k, \{x\}\}$, where $\mathcal{C}_k = \mathcal{C}'_k \cup \{x\}$. $\mathcal{T}'$ and $\mathcal{T}$ are encoding tree according to $\mathcal{P}'$ and $\mathcal{P}$, $\lambda$ is the root node of the encoding tree of graph $G$.

$$
\begin{aligned}
\Delta_L(x, \mathcal{C}) = & \mathcal{H}^2(\mathcal{P}) - \mathcal{H}^2(\mathcal{P}') \\
= & -\sum_{\alpha \in \mathcal{T}, \alpha \neq \lambda} \frac{g_\alpha}{v_\lambda} \log \frac{v_\alpha}{v_{\bar{\alpha}}} + \sum_{\alpha' \in \mathcal{T}', \alpha' \neq \lambda} \frac{g_{\alpha'}}{v_\lambda} \log \frac{v_{\alpha'}}{v_{\bar{\alpha}'}} \\
= & \underbrace{-\frac{g_{c_k}}{v_\lambda} \log \frac{v_{c_k}}{v_\lambda} - \sum_{x_i \in \mathcal{C}_k \backslash \{x\}} \frac{d_{x_i}}{v_\lambda} \log \frac{d_{x_i}}{v_{c_k}}}_{\mathcal{H}^2(\mathcal{C}_k \backslash \{x\})} \underbrace{-\frac{d_x}{v_\lambda} \log \frac{d_x}{v_{c_k}}}_{\mathcal{H}^2(x)} \\
& \underbrace{+\frac{g_{c'_k}}{v_\lambda} \log \frac{v_{c'_k}}{v_\lambda} + \sum_{x_i \in \mathcal{C}'_k} \frac{d_{x_i}}{v_\lambda} \log \frac{d_{x_i}}{v_{c'_k}}}_{-\mathcal{H}^2(\mathcal{C}'_k)} \underbrace{+\frac{d_x}{v_\lambda} \log \frac{d_x}{v_\lambda}}_{-\mathcal{H}^2(\{x\})}
\end{aligned}
$$

$$
\begin{aligned}
= & \frac{g_{c'_k}}{v_\lambda} \log \frac{v_{c'_k}}{v_\lambda} - \frac{g_{c_k}}{v_\lambda} \log \frac{v_{c_k}}{v_\lambda} + \frac{d_x}{v_\lambda} (\log \frac{d_x}{v_\lambda} - \log \frac{d_x}{v_{c_k}}) \\
& + \sum_{x_i \in \mathcal{C}'_k} \frac{d_{x_i}}{v_\lambda} (\log \frac{d_{x_i}}{v_{c'_k}} - \log \frac{d_{x_i}}{v_{c_k}}) \\
= & \frac{g_{c'_k}}{v_\lambda} \log \frac{v_{c'_k}}{v_\lambda} - \frac{g_{c_k}}{v_\lambda} \log \frac{v_{c_k}}{v_\lambda} + \frac{d_x}{v_\lambda} \log \frac{v_{c_k}}{v_\lambda} \\
& + \log \frac{v_{c_k}}{v_{c'_k}} \sum_{x_i \in \mathcal{C}'_k} \frac{d_{x_i}}{v_\lambda} \\
= & \frac{g_{c'_k}}{v_\lambda} \log \frac{v_{c'_k}}{v_\lambda} - \frac{g_{c_k}}{v_\lambda} \log \frac{v_{c_k}}{v_\lambda} + \frac{d_x}{v_\lambda} \log \frac{v_{c_k}}{v_\lambda} + \frac{v_{c'_k}}{v_\lambda} \log \frac{v_{c_k}}{v_{c'_k}}
\end{aligned} \tag{17}
$$

where, $d_x^{\text{in}}$ denotes the sum of edges from node $x$ to nodes $x_j \in \mathcal{C}_k$. Since the $v_{c_k}$, $g_{c_k}$ and $d_x^{\text{in}}$ can be cached during the algorithm iterations:

$$
v_{c'_k} = v_{c_k} - d_x, \quad g_{c'_k} = g_{c_k} - 2 * d_x^{\text{in}} + d_x. \tag{18}
$$

# C EVALUATION METRICS.

Given a ground truth community $\mathcal{C} \subseteq \mathcal{V}$ and a reconstructed community $\mathcal{C}' \subseteq \mathcal{V}$, the precision $P(\mathcal{C}', \mathcal{C})$ and recall $R(\mathcal{C}', \mathcal{C})$ are defined as follows:

$$
P(\mathcal{C}', \mathcal{C}) = \frac{|\mathcal{C} \cap \mathcal{C}'|}{|\mathcal{C}'|}, \quad R(\mathcal{C}', \mathcal{C}) = \frac{|\mathcal{C} \cap \mathcal{C}'|}{|\mathcal{C}|}. \tag{19}
$$

Precision represents the fraction of the reconstruction in the ground truth, while recall represents the fraction of the ground truth in the reconstruction. These notions are often combined into a single number between 0 and 1, known as the $F_1$-score, defined as:

$$
F_1(\mathcal{C}', \mathcal{C}) = 2 \cdot \frac{P(\mathcal{C}', \mathcal{C}) \cdot R(\mathcal{C}', \mathcal{C})}{P(\mathcal{C}', \mathcal{C}) + R(\mathcal{C}', \mathcal{C})}. \tag{20}
$$

The $F_1$-score has the additional advantage of being symmetric, i.e., $F_1(\mathcal{C}', \mathcal{C}) = F_1(\mathcal{C}, \mathcal{C}')$, and it equals 1 if and only if the sets $\mathcal{C}$ and $\mathcal{C}'$ are identical.

To evaluate the set of detected clusters, we define the $F_1$ score for a collection of ground truth communities $\mathcal{P}$ and a collection of detected communities $\mathcal{P}'$ [30, 50]. The F1-score for detection is calculated as the average of two values: the F1-score of the best-matching ground-truth community for each detected community, and the F1-score of the best-matching detected community for each ground-truth community:

$$
F_1(\mathcal{P}', \mathcal{P}) = \frac{1}{2} \left( \frac{1}{|\mathcal{P}'|} \sum_{\mathcal{C}' \in \mathcal{P}'} \max_{\mathcal{C} \in \mathcal{P}} F_1(\mathcal{C}', \mathcal{C}) + \frac{1}{|\mathcal{P}|} \sum_{\mathcal{C} \in \mathcal{P}} \max_{\mathcal{C}' \in \mathcal{P}'} F_1(\mathcal{C}, \mathcal{C}') \right). \tag{21}
$$

We also use the Overlapping Normalized Mutual Information (ONMI) based on information theory developed by Lancichinetti and Fortunato [24] and later refined by McDaid et al. [35].

$$
\text{ONMI}(\mathcal{P}', \mathcal{P}) = \frac{I(\mathcal{P}', \mathcal{P})}{\max(H(\mathcal{P}'), H(\mathcal{P}))}, \tag{22}
$$

where $H(\mathcal{P})$ is the Shannon entropy of $\mathcal{P}$, and $I(\mathcal{P}', \mathcal{P})$ is the mutual information. It is important to note that the difference between

NMI and ONMI lies in that nodes in $\mathcal{P}$ and $\mathcal{P}'$ only belong to a single community. For the specific calculation formula, refer to literature [35]. F1 measures the node-level detection performance, while NMI aims at the community-level detection performance. These values are in [0, 1], with 1 representing perfect matching.

## D DATASETS

We conduct comprehensive experiments on nine real-world, large-scale networks, which are described in detail as follows:

- **Amazon**. The Amazon network is constructed by crawling the Amazon website. If a product $i$ is frequently co-purchased with product $j$, the graph contains an undirected edge from $i$ to $j$. Each product category provided by Amazon defines a ground-truth community.
- **YouTube**. YouTube is a video-sharing website that includes a social network where users can form friendships and create groups that other users can join. These user-defined groups are considered ground-truth communities.
- **DBLP**. DBLP is a co-authorship network where two authors are connected if they have published at least one paper together. Publication venues, such as journals or conferences, define individual ground-truth communities; Authors who have published in a specific journal or conference form a community.
- **LiveJournal**. LiveJournal is a free online blogging community where users can declare friendships with each other. User-defined friendship groups are considered ground-truth communities.
- **Orkut**. Orkut is a free online social network where users can form friendships and create groups that other members can join. These user-defined groups are considered ground-truth communities.
- **Friendster**. Friendster is originally a social networking site where users can form friendships and create groups. It later transitioned into an online gaming platform. User-defined groups are considered ground-truth communities
- **Wiki**. Wiki is a directed overlapping network constructed based on Wikipedia's hyperlink data from September 2011. This network includes page names and corresponding category information, where each article may belong to multiple categories. These categories can be considered as ground truth community labels.
- **Tweet12**. After filtering out duplicates and unusable tweets, the dataset comprises 68,841 English tweets published over four weeks in 2012, covering 503 distinct event types. In the constructed Tweet12 network, each node represents a tweet. We utilize SBERT [41] to embed all tweets' content and calculate the cosine similarity between each pair of tweets. For each node, the top 150 most semantically similar nodes are selected as neighbors. Additionally, nodes are connected based on common attributes, such as hashtags or users. The weight of each edge corresponds to the cosine similarity of the embeddings between the connected nodes.
- **Tweet18**. After filtering out duplicates and unusable tweets, the dataset contains 64,516 French tweets published over 23 days in 2018, covering 257 types of events. The construction method for the social network graph of Tweet18 is identical to that of Tweet12.

## E BASELINES

We compare CoDeSEG with four overlapping community detection algorithms and four non-overlapping community detection algorithms. The detailed descriptions of all baselines are as follows:

- **SLPA** is an overlapping community detection method based on label propagation, designed to identify community structures in networks by propagating labels between nodes. We implement this algorithm in Python3.8 with the NetworkX package in CDlib, with the number of iterations set to 21 and the filtering threshold set to 0.01. The original code is available at https://github.com/kbalasu/SLPA.
- **Bigclam** is an overlapping community detection method for large networks based on Non-negative Matrix Factorization (NMF), which maximizes the likelihood estimate of the graph to find the optimal community structure. We implement this algorithm using open-source code, with the number of communities set to 25,000. The code is available at https://github.com/snap-stanford/snap.
- **NcGame** is an overlapping community detection algorithm based on non-cooperative game theory, which treats nodes as rational, self-interested participants and aims to find communities that maximize individual node benefits. We implement this algorithm with python3.8 via open-source code available in the paper's supplementary material https://doi.org/10.1038/s41598-022-15095-9.
- **Fox** is a heuristic overlapping community detection method that measures the closeness between nodes and communities by approximating the number of triangles formed by nodes and communities. We implement this algorithm using open-source code available at https://github.com/timgarrels/LazyFox, with the stopping threshold set to 0.1.
- **Louvain** is a community detection method based on modularity optimization, which identifies non-overlapping community structures in networks through a multi-level optimization strategy. We implement this algorithm in Python3.8 with API provided by igraph library, and its open-source code is available at https://github.com/taynaud/python-louvain.
- **DER** is a diffusion entropy reduction graph clustering algorithm with random walks and k-means for non-overlapping community detection. We implement this algorithm in Python3.8 with the NetworkX package in CDlib, and its original code is available at https://github.com/komarkdev/der_graph_clustering.
- **Leiden** is an improved version of the Louvain algorithm, which enhances community detection accuracy and stability through local optimization and refinement steps. We implement this algorithm using open-source code available at https://github.com/vtraag/leidenalg.
- **FLPA** is a fast variant of LPA [39] that is based on processing a queue of nodes whose neighborhood recently changed. We implement this algorithm in Python3.8 with API provided by igraph library, and its open-source code is available at https://github.com/vtraag/igraph/tree/flpa.

## F VISUALIZATION

To intuitively assess the quality of community detection across various algorithms, we visualize the network consisting of the top 500

(a) Ground truth.            (b) CoDeSEG.            (c) Fox.

(d) NcGame.            (e) SLPA.            (f) Bigclam.

**Figure 6: Community detection result visualization on Amazon (top 500 communities). Light gray indicates community nodes ranked beyond 100, while gray denotes nodes misaligned with the ground truth.**

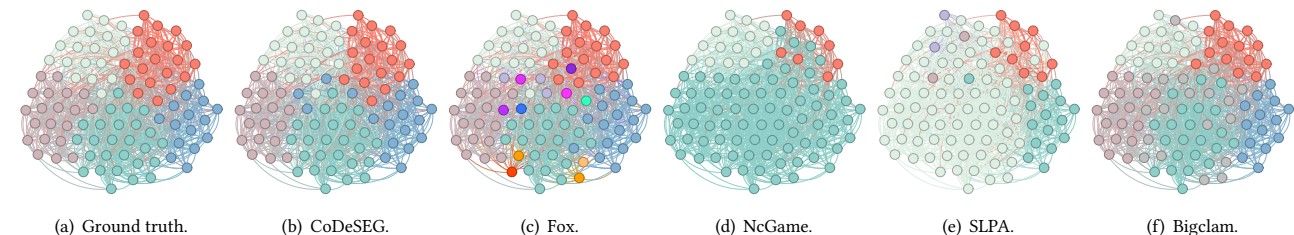

(a) Ground truth.    (b) CoDeSEG.    (c) Fox.    (d) NcGame.    (e) SLPA.    (f) Bigclam.

**Figure 7: Community detection result visualization on LFR benchmark network.**

communities by node count from the Amazon dataset, comprising 13,132 nodes and 41,263 edges. For clarity, the top 100 communities are highlighted with distinct colors. As illustrated in Figure 6, the detected communities by the proposed CoDeSEG closely align with baselines NcGame and SLPA, exhibiting the highest similarity to the ground truth. In contrast, the results of Fox and Bigclam display a higher proportion of gray nodes ( i.e., nodes that do not match ground truth assignments), with Fox showing the most significant deviation from the ground truth communities. To clearly demonstrate the detection performance of the CoDeSEG algorithm, we visualize the detection results of various algorithms on the LFR benchmark network [25]. The constructed LFR benchmark network

comprises five communities, 100 nodes, and 977 edges. The visualization in Figure 7 demonstrates that the detected communities by CoDeSEG are almost identical to the ground truth. While Fox accurately assigns most nodes to correct communities, it detects more communities than the ground truth. NcGame and SLPA, on the other hand, tend to assign most nodes to fewer communities. Bigclam, in contrast, fails to assign some nodes to any community (i.e., the gray nodes in Figure 7(f)), which are considered noise or not part of any significant community. Overall, CoDeSEG consistently delivers superior performance across different networks, demonstrating enhanced stability and robustness.

