# OpenReview forum: "Community Detection in Large-Scale Complex Networks via Structural Entropy Game"
_ACM.org/TheWebConf/2025/Conference — WWW 2025 Poster_

### Official Review · Reviewer_L8AS · 2024-11-30

**Novelty:** 3
**Technical Quality:** 4

**Review:**

Summary:

The paper introduces CoDeSEG, a novel game-theoretic framework for community detection in large-scale networks, utilizing a structural entropy-based approach. The method defines a two-dimensional structural entropy where nodes iteratively adjust their community memberships to minimize this entropy. This heuristic strategy ensures efficient detection of both non-overlapping and overlapping communities, even in networks with millions of nodes and edges.


Strengths:

1. The method game-theoretic framework by leveraging structural entropy to achieve nearly linear time complexity, ensuring scalability for large-scale networks.
2. The algorithm operates efficiently on both weighted and unweighted networks, outperforming existing methods in speed and accuracy


Weaknesses:

1. Limited novelty in the proposed methodology. The paper does not clearly articulate the rationale behind the newly design of the structural entropy or its physical interpretation, leaving its conceptual foundation ambiguous.
2. The relationship between the structural entropy design and its ability to address challenges in unweighted or large-scale networks is not explicitly established, making its practical relevance unclear.

**Questions:**

1. The rationale for the design of structural entropy is unclear. How does the newly designed structural entropy effectively measure uncertainty and information content in complex networks? A clearer explanation of its theoretical foundation would strengthen the paper.

2. The physical meaning of structural entropy is missing. How do high or low values correlate with the quality of community assignments? Additionally, how does the proposed 2D structural entropy serve as a superior metric compared to the original structural entropy for community detection? The related analysis and ablation study are necessary.

3. The relationship between the design of structural entropy and its applicability to challenges in large-scale and unweighted networks is not well established. How does the design of structural entropy address the problems of community detection on larger-scale networks and unweighted networks?

4. Further clarification is needed to highlight the contribution of CoDeSEG, particularly in comparison to prior works, such as FOX, Louvain. What specific improvements or innovations does CoDeSEG offer over these methods in terms of solving community detection in large-scale networks?

5. The setting for community detection in weighted networks is missing. Could the authors elaborate on this aspect and explain the relationship between the structural entropy design and its capability to address community detection in weighted networks?

6. Why is the computation of ``leave and be alone” performed within the loop for the 𝑘-th adjacent community in line 13 of Algorithm 1? Additionally, how is a neighboring community defined if a node is located within the community (rather than on its boundary)?

**Reviewer Confidence:**

4: The reviewer is certain that the evaluation is correct and very familiar with the relevant literature

**Scope:**

4: The work is relevant to the Web and to the track, and is of broad interest to the community

---

### Official Review · Reviewer_27Ru · 2024-11-30

**Novelty:** 5
**Technical Quality:** 5

**Review:**

This paper designs a greedy community detection algorithm by combining latent game theory with the minimization of network structural entropy. This is a novel approach, and it achieves community detection with low time complexity in offline scenarios, though it cannot guarantee high accuracy. Additionally, the authors have considered non-overlapping community detection.

Some interesting points:

1. The use of game theory and structural entropy for community detection.

2. Low computational time complexity.

3. Provides a parallel computing approach.

Some concerns:

1. The determination of overlapping communities is based on non-overlapping communities but does not affect the latter. This seems counterintuitive, as in reality, overlapping community detection should be holistic and more dynamic compared to non-overlapping detection.

2. Initializing with n communities is not a reasonable approach, and it raises concerns about potential high memory overhead.

3. The experiments presented are somewhat confusing. For example, in Figure 4, both \tau_n and the number of iterations change simultaneously without being separately addressed.

4. The writing needs further refinement, especially regarding the algorithm. Some parameters should be defined as inputs.

**Questions:**

a) In the process of detecting overlapping communities, the paper seems to consistently use non-overlapping communities as the standard. After identifying overlapping communities, the structure of the original communities should have changed. Would this impact the validity of the authors' results?

b) The writing of the paper needs to be restructured, especially the algorithm section. Certain parameters (e.g., thresholds \tau_n and \tau_o are neither initialized nor presented as input parameters.

c) As mentioned in (a), the parallel strategy relies on the results of non-overlapping community detection. During the overlapping community detection phase, maintaining unchanged non-overlapping community detection results introduces a counterfactual inconsistency.

d) A calculation of the space complexity needs to be provided.

e) Figure 4 is rather confusing. Why are both the threshold \tau_n and the number of iterations being changed simultaneously? It is unclear what the authors aim to convey. Additionally, why did the authors not analyze \tau_o?

f) Regarding the iterative training process, how were the training parameters configured by the authors?

**Reviewer Confidence:**

3: The reviewer is confident but not certain that the evaluation is correct

**Scope:**

4: The work is relevant to the Web and to the track, and is of broad interest to the community

---

### Official Review · Reviewer_ccuq · 2024-12-01

**Novelty:** 5
**Technical Quality:** 5

**Review:**

Summary: The paper proposes a novel Community Detection in Large-Scale Complex Networks via Structural Entropy Game (CoDeSEG), inspired by game theory, which minimizes two-dimensional structural entropy to identify communities in large-scale networks. The algorithm runs with near-linear time complexity and achieves state-of-the-art performance in terms of ONMI and F1 score on real-world datasets.

Strengths: 1) This paper effectively addresses the challenge of detecting overlapping communities in large-scale networks by utilizing two-dimensional structural entropy and the concept of community formation as a game-theoretic process. 2)The details of the paper are well grasped, with thorough explanations of the formulas, algorithms, datasets, and experiments

Weaknesses: 1) The concept of Nash equilibrium is critical to the application of structural entropy and game theory, yet the manuscript does not provide a clear explanation. It is recommended to add a brief explanation of Nash equilibrium, including its relevance to the context of the study. For clarity, consider including the standard mathematical definition or illustrative formulas to enhance the reader's understanding. 2) The proposed method follows the game-theoretic inspired
community detection framework  [9], but the experiments fail to include a direct comparison with [9]. Including results from [9] as a baseline would significantly enhance the manuscript's credibility and provide readers with a clearer perspective on the contributions. 3) Section 4.2 dedicates excessive space to presenting experimental results, but lacks substantial analysis or interpretation.

**Questions:**

If Eq. 5 equals zero, does it mean that the node may be in the "stay" or "leave and be alone" state? How does this setting affect the results?

**Reviewer Confidence:**

4: The reviewer is certain that the evaluation is correct and very familiar with the relevant literature

**Scope:**

4: The work is relevant to the Web and to the track, and is of broad interest to the community

---

### Official Review · Reviewer_teHj · 2024-12-01

**Novelty:** 4
**Technical Quality:** 5

**Review:**

The paper proposes a novel heuristic community detection algorithm inspired by game theory, termed CoDeSEG, which can apply to various graphs, whether unweighted, weighted, undirected, or directed graphs, to produce stable, reliable community structures in a unified framework.

Strengths:
1. The proposed method can apply to various graphs, whether unweighted, weighted, undirected, or directed graphs, to produce stable, reliable community structures in a unified framework.
2. Clear structure, logical flow, and ease of understanding.

Weaknesses:
1. The framework provided in the manuscript does not sufficiently emphasize the novel contributions of the study. It is better to revise the framework to explicitly showcase the innovative aspects of the proposed model.
2. The manuscript lacks a definition of Nash equilibrium.
3. The manuscript asserts that the proposed method achieves state-of-the-art performance. However, most of the comparison algorithms are outdated, with the latest ones published in 2023 and 2022, and others dating back to before 2020. It is better to include comparisons with more recent algorithms, particularly those published in 2024.

**Questions:**

1. What is the motivation for selecting ONMI, NMI, and F1-Score as evaluation metrics? How does the algorithm perform when (overlapping) modularity is used as the evaluation metric?
2. If the real (overlapping) communities of the test networks are known in advance as the authors use the criterion NMI (ONMI) and F1-Score? In this case, please test certain real-world networks without knowing the real communities. In this case, how to evaluate the performance of the proposed method?

**Reviewer Confidence:**

3: The reviewer is confident but not certain that the evaluation is correct

**Scope:**

4: The work is relevant to the Web and to the track, and is of broad interest to the community

---

### Official Review · Reviewer_L1Mq · 2024-12-03

**Novelty:** 5
**Technical Quality:** 6

**Review:**

## Quality
The paper presents a novel algorithm, CoDeSEG (Community Detection via Structural Entropy Game), designed for detecting communities in large-scale networks. It introduces significant improvements in both efficiency and accuracy for overlapping and non-overlapping community detection tasks. The methodology, combining structural entropy minimization with game-theoretic principles, is well-motivated and supported by extensive experimental validation.

## Clarity
The paper is generally well-structured and clearly written. The concepts of structural entropy, game theory, and their application to community detection are explained in detail, accompanied by mathematical formulations. However, there are areas where the presentation could be improved. For instance, the clarity of Figure 3 is slightly compromised.

## Originality
The proposed method demonstrates originality by integrating two-dimensional structural entropy as a utility function within a potential game framework. The innovative use of structural entropy for defining overlapping community heuristics is particularly noteworthy. Compared to traditional approaches like modularity maximization or label propagation, CoDeSEG introduces fresh perspectives.

## Significance
The algorithm's demonstrated scalability (linear time complexity) and robustness across diverse network types (weighted, directed, undirected) make it a valuable contribution to the field of network science. The ability to handle large-scale datasets efficiently while achieving state-of-the-art performance underscores its practical significance.

## Pros
- Innovative Approach: Novel integration of structural entropy and potential games for community detection.
- Scalability: Demonstrated capability to handle millions of nodes in real-world networks.
- Flexibility: Applicability to various graph types, including weighted and directed networks.
- Efficiency: Achieves significant speedup compared to baseline methods, with convergence in a few iterations.
- Comprehensive Validation: Experimental results on multiple large-scale datasets, with metrics like ONMI and F1 score.

## Cons
- Complexity of Understanding: Some mathematical notations and derivations are dense, potentially hindering comprehension for broader audiences.
- Overlapping Detection Heuristics: While innovative, the overlap criteria could lead to excessive or inadequate overlaps without precise parameter tuning.

**Questions:**

- Termination Threshold: The time consumption is also related to $𝜏_𝑛$. Have you tested the effect of the $𝜏_𝑛$ parameter on datasets other than Amazon and DBLP?
- Heuristic Sensitivity: How sensitive is the algorithm's performance to the chosen thresholds for overlapping (e.g., $𝜏_o$​) and non-overlapping communities? Could you provide guidelines for their optimal selection?
- Failure Cases: Are there specific network structures or scenarios where CoDeSEG performs poorly compared to other baselines? If yes, what are potential remedies?

**Reviewer Confidence:**

3: The reviewer is confident but not certain that the evaluation is correct

**Scope:**

4: The work is relevant to the Web and to the track, and is of broad interest to the community